# Monitoring and Evaluation of National Vaccination Implementation: A Scoping Review of How Frameworks and Indicators Are Used in the Public Health Literature

**DOI:** 10.3390/vaccines10040567

**Published:** 2022-04-06

**Authors:** Manar Marzouk, Maryam Omar, Kanchanok Sirison, Aparna Ananthakrishnan, Anna Durrance-Bagale, Chatkamol Pheerapanyawaranun, Charatpol Porncharoen, Nopphadol Pimsarn, Sze Tung Lam, Mengieng Ung, Zeenathnisa Mougammadou Aribou, Saudamini V. Dabak, Wanrudee Isaranuwatchai, Natasha Howard

**Affiliations:** 1Saw Swee Hock School of Public Health, National University of Singapore, National University Health System, 12 Science Drive 2, Singapore 117549, Singapore; manar.marzouk@visitor.nus.edu.sg (M.M.); st.lam@u.nus.edu (S.T.L.); mung@nus.edu.sg (M.U.); e0431577@u.nus.edu (Z.M.A.); natasha.howard@nus.edu.sg (N.H.); 2Barts Health NHS Trust, Newham University Hospital, London E13 8SL, UK; maryam.omar@doctors.org.uk; 3Health Intervention and Technology Assessment Program, Ministry of Public Health, Nonthaburi 11000, Thailand; aparna.ananthakrishnan@gmail.com (A.A.); chatkamol.ph@outlook.co.th (C.P.); charatpol.por@g.swu.ac.th (C.P.); nopphadol.p@hitap.net (N.P.); saudamini.d@hitap.net (S.V.D.); wanrudee.i@hitap.net (W.I.); 4London School of Hygiene and Tropical Medicine, 15-17 Tavistock Place, London WC1H 9SH, UK; 5Institute of Health Policy, Management and Evaluation, University of Toronto, 155 College St., Toronto, ON M5T 3M6, Canada

**Keywords:** vaccination, monitoring, evaluation, indicators, global health

## Abstract

An effective Monitoring and Evaluation (M&E) framework helps vaccination programme managers determine progress and effectiveness for agreed indicators against clear benchmarks and targets. We aimed to identify the literature on M&E frameworks and indicators used in national vaccination programmes and synthesise approaches and lessons to inform development of future frameworks. We conducted a scoping review using Arksey and O’Malley’s six-stage framework to identify and synthesise sources on monitoring or evaluation of national vaccination implementation that described a framework or indicators. The findings were summarised thematically. We included 43 eligible sources of 4291 screened. Most (95%) were in English and discussed high-income (51%) or middle-income (30%) settings, with 13 in Europe (30%), 10 in Asia-Pacific (23%), nine in Africa (21%), and eight in the Americas (19%), respectively, while three crossed regions. Only five (12%) specified the use of an M&E framework. Most (32/43; 74%) explicitly or implicitly included vaccine coverage indicators, followed by 12 including operational (28%), five including clinical (12%), and two including cost indicators (5%). The use of M&E frameworks was seldom explicit or clearly defined in our sources, with indicators rarely fully defined or benchmarked against targets. Sources focused on ways to improve vaccination programmes without explicitly considering ways to improve assessment. Literature on M&E framework and indicator use in national vaccination programmes is limited and focused on routine childhood vaccination. Therefore, documentation of more experiences and lessons is needed to better inform vaccination M&E beyond childhood.

## 1. Introduction

Improving national vaccination programme implementation requires collection and analysis of data on relevant vaccination components. Monitoring and evaluation (M&E) or more recent Monitoring, Evaluation, Accountability and Learning (MEAL) frameworks [1] support decision-making by consolidating available information on agreed indicators, benchmarked targets, and methods to collect, analyse, and report necessary data to strengthen vaccination programmes [2]. M&E frameworks are usually aggregated into pre-, peri-, and post-vaccination phases, and include elements of vaccine procurement, transport, storage, staff training, communication, coverage, adverse effects, and identification of successes and failures [3].

Planning effective M&E for national rollout of new vaccines, such as for COVID-19, can be strengthened by learning from previous vaccination experiences, particularly those targeted beyond routine childhood populations [4]. A virtual expert roundtable, hosted by the Saw Swee Hock School of Public Health in January 2021, identified key M&E framework components to inform COVID-19 vaccination. This included best practice guidelines, particularly by the World Health Organization (WHO) [5], but few lessons or experiences of using M&E frameworks and selecting and appropriately benchmarking indicators within vaccination programme M&E. Practical details of these experiences could help governments and technical partners in planning, implementing, and assessing M&E for new vaccine implementation. Lessons learnt from assessment experiences worldwide could help inform national efforts to improve routine and vaccine-specific data collection and analyses and support vaccination programme strengthening, particularly in resource-constrained settings, which aligns with the Immunization Agenda 2030 goal to make vaccination available to everyone, everywhere [6].

We thus aimed to synthesise the literature on M&E frameworks and indicators used for vaccination implementation. The objectives were to: (i) summarise the scope of existing primary literature on M&E frameworks or indicators used; (ii) identify any useful indicators to inform development or adaptation of M&E frameworks; and (iii) synthesise lessons to inform M&E framework development for national rollouts of vaccination.

## 2. Materials and Methods

### 2.1. Study Design

We conducted a scoping literature review using Arksey and O’Malley’s six-stage framework with Levac and colleagues’ revisions and Khalil and colleagues’ refinements [7,8,9]. We selected this method because, as Munn et al. suggested, scoping reviews are useful to map and identify evidence in emerging topics and help identify key concepts and gaps [10]. 

### 2.2. Identifying the Research Question (Stage 1)

Our research question was: “What are the scope (i.e., extent, distribution, nature), main findings, and key lessons of literature on M&E frameworks and indicators for vaccination implementation?” Table 1 provides our study definitions. 

### 2.3. Identifying Relevant Sources (Stage 2)

To ensure breadth, we included multiple electronic databases and websites. First, we searched five databases systematically (i.e., Medline, Embase, Web of Science, Scopus, Eldis). Second, we searched eight relevant websites purposively (i.e., WHO, Australian Department of Health, National Advisory Committee on Immunization Canada, India Ministry of Health and Family Welfare, Philippines Department of Health, Singapore Ministry of Health, UK Joint Committee on Vaccination and Immunisation, US Centers for Disease Control and Prevention). For both databases and websites, we used search terms for ‘vaccine’ (i.e., vaccin*, immuniz*, immunis*) AND ‘monitoring’ AND ‘evaluation’ (i.e., monitor* and evaluat*, M&E, Monitor*, evaluat*) and related terminology adapted to subject headings. 

### 2.4. Selecting Sources (Stage 3)

We established eligibility criteria based on our research question and discussion with experts (Table 2). We included primary research sources focused on vaccine implementation in national settings and including content on M&E frameworks or indicators. Thus, we also included conference abstracts, commentaries, book chapters, and reviews that provided research data not already included in a research article. We did not exclude on language (if an English abstract was accessible), publication year, study design, or participants. 

We screened 4288 potential sources using Covidence and EndNote software. After removing 2089 duplicates, all authors first screened 2199 titles and abstracts against eligibility criteria and excluded 1995 ineligible sources. We then screened 204 full texts, excluding another 163 ineligible sources. We added two eligible website sources to 41 eligible database sources, thus including 43 in total.

### 2.5. Extracting (Charting) Data (Stage 4)

We extracted data from 43 sources into Excel using the following headings: lead author, publication year, source type (i.e., article, abstract, book, report), language, country/ies included, aim, study information (i.e., design, participants, data collection, analysis), and findings (i.e., M&E tool/framework used, indicators included, lessons described). Indicators were subcategorised as coverage (i.e., targeting, population estimation, equity, disaggregation, uptake/coverage, attitude/behaviours), operational (i.e., health service capacity, human resources, vaccine supply chain (e.g., availability, allocation, transport, storage, delivery, wastage, disposal)), clinical (i.e., vaccine safety, vaccine demand), or others (i.e., costs, additional indicators) as described in the WHO-UNICEF monitoring framework for COVID-19 vaccines [12]. 

### 2.6. Collating and Summarising Findings (Stage 5)

First, we summarised sources extent (i.e., database/website origin, publication year), distribution (i.e., publication language, countries included), and nature (i.e., type, topic, study design, outcomes included). Second, we synthesised data thematically under framework and indicator headings guided by our research question and stakeholder consultation. 

### 2.7. Consulting Stakeholders (Stage 6)

We discussed initial review findings with 16 high-level stakeholders in the Thai Ministry of Public Health in December 2021. Stakeholders were asked how findings could be made most useful and for any additional potential sources (none were identified). Inputs informed final synthesis.

## 3. Results

### 3.1. Scope of the Literature

Figure 1 provides the PRISMA flow diagram of the 43 eligible sources of 4291 screened. Databases provided 4288 (i.e., 1122 in Medline, 1746 in Embase, 600 in Web of Science, 820 in Scopus, 0 in Eldis) and the UK Joint Committee on Vaccination and Immunisation website provided two [13,14]. 

Figure 2 shows the extent of sources by publication year. None were found before 1987 or in 1991–2005. From 2006, publications slowly increased, with two notable increases in 2010 and 2012, to a peak of nine in 2018–2019, and then decreased.

Forty single-country sources were distributed across 26 countries, while three multi-country sources included Bahrain, Kuwait, Oman, Qatar, Saudi Arabia, United Arab Emirates [15]; China, Indonesia, Viet Nam [16]; and Bangladesh, Mozambique, Uganda, and Zambia [17]. Over half of sources described high-income settings (22/43; 51%), while 13 (30%) described middle-income settings and only 8 (19%) described low-income settings. 

Most (41/43; 95%) were published in English, with one each in French and Spanish. Most (40/43; 93%) were journal articles, and three (7%) were conference abstracts. Study designs and methodology were often unclear, but 34 sources (79%) appeared to use primarily quantitative, six (14%) used mixed-method, and three (7%) used qualitative approaches. Methods were somewhat better described and included surveys, document analysis, observations, interviews, and focus group discussions.

### 3.2. Synthesised Findings

We synthesised findings under: (i) description of any framework and how it was used; (ii) coverage indicators; (iii) operational indicators; and (iv) clinical indicators. As most sources did not detail the specific frameworks or indicators used, we instead reported on ways they were used and any lessons within each sub-section (Table 3).

#### 3.2.1. Frameworks

Five (12%) sources explicitly described using any type of M&E framework, while the rest either did not use one or were unclear about whether or how one was used and what it included. Thus, framework usage was minimally described and heterogeneous, depending on requirements and objectives. For example, Al Awaidy et al. used the WHO M&E framework for hepatitis B in reviewing vaccination in several Gulf countries [15]. Dang et al. used the mHealth Assessment and Planning for Scale (MAPS) toolkit to assess scale-up of an electronic immunisation registry in Viet Nam [18]. Hutubessy et al. used the Cervical Cancer Prevention and Control Costing (C4P) tool to determine cost-effectiveness of HPV vaccination in Tanzania [19]. Ijsselmuiden et al. used the WHO Extended Programme on Immunisation framework to determine vaccination coverage and cold chain maintenance in South Africa [20]. Non-WHO M&E frameworks included Manyazewal and colleagues’ use of the Plan-Do-Check-Act (PDCA) cycle with Continuous Quality Improvement, a prospective quasi-experimental interrupted time-series design to evaluate effectiveness of a continuous quality improvement intervention for a vaccination programme in Ethiopia [21] and Aceituno and colleagues’ use of a logical framework to determine participant engagement in Bolivia [22]. 

Four (9%) sources did not specify the use of a formal framework, instead describing an assessment process or method (e.g., use of registers, sampling approaches, surveys). For example, Lanata et al. used lot quality assurance sampling to determine vaccine coverage in Peru [23], and Tuells et al. used a WHO process for cold chain temperature monitoring in Spain [24]. Sources seldom explicitly distinguished between routine data collection sources (e.g., national infectious disease surveillance), dedicated disease-specific registers (e.g., for rabies, tetanus, HIV), general surveys (e.g., national-level demographic and health surveys or multiple-indicator cluster surveys), or vaccine-specific (e.g., post-introduction evaluation surveys).

General lessons included incorporating national staff in monitoring meetings to improve M&E ownership and accountability [22], improving contextual dynamics tailored to the specific vaccination programme to improve coverage [21], and multi-disciplinary co-production and inclusion of M&E staff during decision-making to improve outcomes [17].

#### 3.2.2. Coverage Indicators

Most (32/43; 74%) described elements of target population estimation, equity, and uptake, primarily of routine childhood vaccines. Thirteen (30%) included targeting or population estimation, although most did not describe indicators in depth and how they were used differed by setting and resource availability. For example, Lacapere et al. roughly estimated measles-rubella vaccination coverage by dividing the number of vaccine doses given by the estimated population for each district in Haiti [25], while D’Ancona et al. used an immunisation register to estimate coverage in Italy [26]. Bianco et al. determined the number of foreign workers eligible for vaccination in Italy through screening [27]. Manyazewal et al. used WHO Reaching Every Community (REC) mapping of community locations and characteristics to help estimate coverage targets for five vaccines in Ethiopia [21].

Only three (7%) included indicators to examine equity or disaggregate data. For example, Sarker et al. compared immunisation coverage among children aged 12–59 months in Bangladesh across socioeconomic and demographic factors, finding disparities by parental education and mothers’ access to media [28]. Wattiaux et al. considered equity aspects of vaccination rollout by comparing hepatitis B immunisation incidence between indigenous and non-indigenous Australians [29]. Geoghegan et al. examined whether women received COVID-19 vaccine when pregnant or received routinely recommended vaccines in pregnancy in Ireland [30]. 

Sixteen sources (37%) discussed uptake indicators. Most focused on general target populations with minimal discussion on vaccine coverage for migrants and refugees and none on the elderly, people with disabilities, or other potentially vulnerable groups. However, Bawa et al. estimated oral polio vaccination coverage among underserved hard-to-reach communities in Nigeria, defining them based on difficulty of terrain, any local or state border, scattered households, nomadic, water-logged/riverine, or conflict-affected and thus requiring outreach services [31]. More generally, Lacapere et al. calculated numbers of municipalities that reported achieving 95% vaccination coverage for measles-rubella vaccine in Haiti [25]. Only two sources (7%) described coverage indicators beyond the second year of life. Muhamad et al. calculated total HPV vaccine doses delivered through school-based outreach to evaluate the effectiveness of free vaccination for schoolgirls in Malaysia [32], while Beard et al. estimated 40–50% coverage of pertussis vaccination among pregnant women in Australia, based on number of births and consent forms returned centrally [33]. 

Coverage lessons were varied. Aceituno et al. described challenges of collecting high-quality data in resource-constrained settings [22]. Soi et al. noted that using feedback loops to guide policy decision must be pragmatic, as they are often too slow—e.g., Gavi’s HPV demonstration project policy required countries to demonstrate adequate coverage before applying for rollout funding, which could take years [17]. Alam et al. found automation of EPI scheduling can improve coverage and enhance monitoring, particularly in remote areas [34]. Edelstein et al. found that data triangulation and inclusion of routine data, in countries with good national records, can help identify vulnerable groups and monitoring of vaccine coverage [35]. Lanata et al. found lot quality assurance sampling helped identify small areas with poorer vaccination coverage in rural areas of Peru with dispersed populations, thus improving coverage and equity monitoring [23]. Aceituno et al. found staff understanding of cultural-linguistic context improved coverage and vaccination continuity in Bolivia [22].

#### 3.2.3. Operational Indicators 

Only two (5%) sources mentioned health service capacity indicators. Manyazewal et al. assessed immunisation services availability, regular static immunisation services delivered, adequate outreach sites, catchment area mapped for immunisation, separate and adequate rooms for immunisation services and storing supplies, all planned outreach sessions conducted, health education on immunisation provided, and immunisation services availability in all catchment health posts in Ethiopia [21]. Walker et al. assessed surveillance feedback reports, timely reporting, and number of districts with populations not receiving immunisation services [36]. 

Three (7%) sources mentioned supply chain and logistics indicators. Walker et al. used cold chain and logistics data from facility inventory logs for routine immunisation to identify gaps in vaccine supplies and equipment, such as the number of facilities with insufficient supply of syringes and diluent, and number of facilities with inventory logs consistent with vaccine supply [36]. Hipgrave et al. reviewed evidence on thermostability of hepatitis B vaccine for pregnant women when stored outside the cold chain in China [16]. Özdemir et al. assessed cold chain storage and gaps for a hepatitis B vaccine in Turkey [37]. Manyazewal et al. assessed adequacy of fridge-tag 2 units for temperature monitoring, refrigerator spare parts, vaccine request and report forms, and inventory documents in Ethiopia [21].

Five (12%) sources mentioned human resource indicators. Hall et al. assessed the number of English health-workers vaccinated against COVID-19 stratified by dose, manufacturer, and day [13]. Cherif et al. assessed numbers of health-workers participating in vaccination activities, e.g., epidemiological surveillance, adverse event monitoring training, and supervisions in Abidjan [38]. Carrico et al. assessed numbers of states mandating health-worker vaccination in the US [39]. Manyazewal et al. assessed numbers of experts assigned for immunisation and numbers of immunisation focal persons to evaluate effectiveness of system-wide continuous quality improvement for national immunisation programme performance [21]. Walker et al. assessed numbers of supervisory visits conducted, documented in writing, surveillance guidelines observed, surveillance discussed at supervisory visits, and if an operational plan was observed [36].

Two (5%) sources included indicators for vaccination costing. Hutubessy et al. calculated the incremental costs to the health system of HPV vaccination for adolescent girls through schools, health facilities, and other outreach strategies in Tanzania [20]. Walker et al. measured the number of districts in Kenya with insufficient financial resources for key surveillance elements for acute flaccid paralysis [36]. 

Multiple sources discussed operational lessons. D’Ancona et al. used an observational survey to show that decentralised health systems such as in Italy could result in fragmented immunisation registries and information flow across regions [26]. Ward et al. described how data improvement teams, allocated to all districts, enhanced the quality of vaccine administrative data in Uganda by helping identify data inaccuracies and providing on-the-job data collection training [40]. Dang et al. used qualitative research to describe an approach to optimise vaccination information in Vietnam, which included establishing a partnership between the Vietnamese Ministry of Health and mobile network operators [17]. Soi et al. described the importance of physically co-locating evaluators from different disciplinary backgrounds, and suggested that including evaluators in decision-making could enrich outcomes [17]. 

#### 3.2.4. Clinical Indicators

Five (12%) sources described clinical indicators, primarily counting numbers of adverse events following immunisation (AEFI). For example, Aceituno et al. assessed monthly reporting of adverse events and severe adverse events, details of any deaths, reasons for all withdrawals, and infant and maternal death rates below Demographic and Health Survey rates for Bolivia [22]. Loughlin et al. conducted a post-marketing evaluation to assess the number of confirmed cases of intussusception or Kawasaki disease among infants who received Rotavirus vaccine in the US compared with historical cohort data from diphtheria-tetanus-acellular pertussis vaccination [41]. 

Lessons were relatively limited. Vivekanandan et al. described the positive role of health-workers in assessing vaccine safety indicators in India [42]. Cherif et al. similarly noted that improving AEFI system performance required improved health-worker training, data analysis, and community engagement [38]. Ijsselmuiden et al. suggested that vaccination targeting at-risk populations, such as for Hepatitis B, should ensure concomitant disease surveillance to reduce morbidity and mortality [20]. Similarly, Beard et al. suggested combining AEFI and syndromic surveillance in emergency departments to monitor numbers of pertussis vaccine adverse events and supplementing maternal influenza vaccination AEFI monitoring with mobile phone text messages in Australia [33]. 

## 4. Discussion

This initial review of the use of M&E frameworks and indicators in vaccination highlights the relatively limited literature on this topic. M&E frameworks are important for consolidating selected indicators and described as essential in the Global Vaccine Action Plan (GVAP) [4], yet their use was seldom explicit or clearly defined in our sources. Most sources described assessment methods (e.g., survey, lot quality assurance) rather than the use of a formal M&E framework, suggesting overreliance on individual methods without the benefit of an overarching assessment framework. Similarly, while indicators were described more frequently, they were rarely fully defined or benchmarked against targets, and sources focused on ways to improve vaccination programmes without explicitly considering ways to improve assessment. Given indicators and benchmarked targets are crucial to national vaccination programme M&E these are noteworthy gaps. 

Limited description of M&E framework or indicator use outside routine childhood vaccination was perhaps unsurprising and some of the lessons identified in our review could inform development of monitoring or evaluation for COVID-19 vaccination, or other vaccines, beyond routine childhood populations [43]. It is worth reiterating that most sources described high- or middle-income settings, and none described the use of M&E frameworks or indicators in fragile or conflict-affected settings despite the risk of poorer routine vaccination coverage, weakened health system responses, and infectious disease outbreaks in these settings. For example, 14 million ‘zero-dose’ children, who did not receive an initial dose of required vaccines, live in conflict-affected African countries [44,45]. The limited documentation of equity indicators was unexpected given that improving equity in immunisation is both essential [46] and expected by donors such as Gavi [47], yet equity monitoring seemed limited and ill defined. Historical and socio-cultural influences and biases can influence the effectiveness of data collection and assessment on equity, and thus vaccination programme success. For instance, if some high-risk cohorts are not considered acceptable or relevant and hence not recorded (e.g., men who have sex with men in Gulf states), accurate and detailed assessment becomes impossible [15]. An approach used by M’Bangombe et al. analysed historical data to augment current data on high-risk populations, expanding the cohort of those identified as being at risk for cholera in Malawi [48]. 

Another surprising gap was the limited documentation of operational indicators, given their importance in effective vaccination management. The Organisation for Economic Co-operation and Development (OECD) report on lessons from government evaluations of COVID-19 responses similarly highlighted gaps in cost and health system indicators [43]. Additionally, we found that sources used different types of routine and ad hoc data, including surveillance, disease registers, household surveys, and vaccine-specific surveys. While this is understandable, depending on setting and programming needs [49], justification was not always explicit. 

Less surprisingly, our review showed a preference for quantitative assessment methods, with only three sources using qualitative and five using mixed methods. However, qualitative and mixed-method social science approaches offer deeper insights into how processes work and are particularly useful for equity analyses. For example, Dutta et al. interviewed vaccination decision-makers in India to examine the importance of engaging with communities to promote health equity, and found this required formulating policies and guidelines that clearly define community engagement and its related evaluation metrics [50]. Qualitative methods can also help amplify perspectives and groups that may be less visible, which can be particularly important in reaching zero-dose children [45]. 

What is often missing in evaluations is the impact of vaccination on the general population, particularly for lower efficacy vaccines (e.g., against malaria, cholera, or influenza) that have relatively low demographic impacts even with high vaccine coverage. It may be worth exploring this demographic impact further, as was observed for smallpox vaccination [51,52]. Therefore, further documentation of assessment methods, experiences, and lessons appears necessary to expand the evidence base and help inform ongoing and future vaccination assessment. 

Several potential limitations should be considered. First, while we included five databases and eight websites, we may still have missed relevant sources. It is likely that much of the research on this topic remains unpublished, as evaluations conducted by non-academic bodies (e.g., government, consultants) may not be in the public domain, though sites such as bioRxiv could be useful for manuscripts awaiting peer review. Second, included sources were not assessed for quality, as the purpose was to scope existing literature, and this would have eliminated too many documents. Third, we excluded sources on vaccine development, so may have missed some that discussed vaccine safety. Fourth, many sources only assessed one or more components of the vaccination programme (e.g., financing, equity, personnel), and thus we did not attempt to make direct comparisons of assessments. Fifth, methods were often insufficiently described, so we chose to categorise broadly as “quantitative, qualitative, mixed-methods” approaches rather than trying to provide more detail. Sixth, we focused on the public health literature as most likely to contain M&E frameworks and indicators, rather than conducting a broader search of various social science literatures. We thus may have missed some qualitative frameworks or indicators, e.g., for vaccine hesitancy. Finally, we chose not to include MEAL framework components for advocacy or learning, which could be relevant for future research. 

Overall, our review identified minimal literature describing M&E frameworks or indicators for use in vaccination programme implementation. Numbers of relevant publications increased during the past decade, particularly after the 2012 Middle East Respiratory Syndrome epidemic, but numbers are still small and focused on high- and middle-income countries. Further research and documentation are therefore needed to identify additional public health lessons. 

## Figures and Tables

**Figure 1 vaccines-10-00567-f001:**
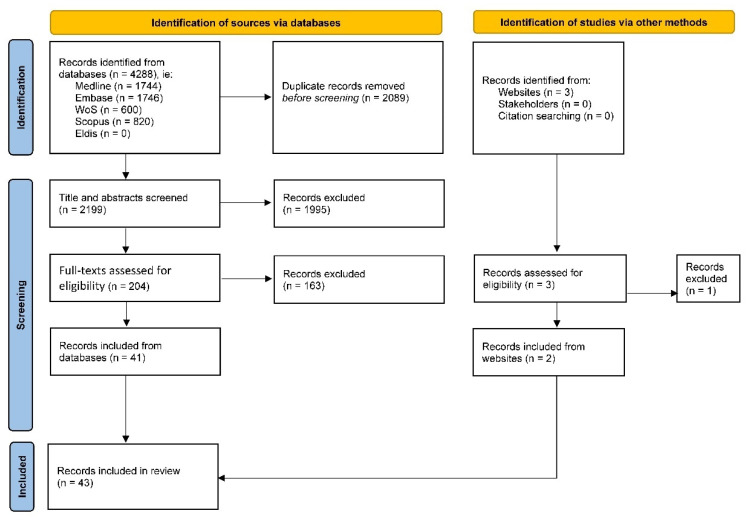
PRISMA flow diagram.

**Figure 2 vaccines-10-00567-f002:**
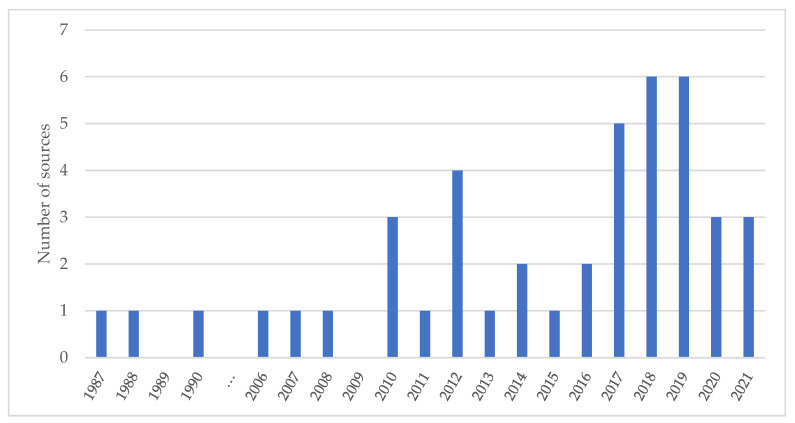
Number of sources by publication year.

**Table 1 vaccines-10-00567-t001:** Study definitions.

Terms	Definitions
Evaluation	The systematic assessment of an activity, project, programme, strategy, policy, topic, theme, sector, operational area or institution’s performance to determine its relevance, effectiveness, efficiency, impact, and/or sustainability [11].
Framework	Shows how the programme or activity is intended to work by organising out the components of the initiative and the order or the steps needed to achieve the desired results. A framework increases understanding of the programme’s goals and objectives, defines the relationships between factors key to implementation, and articulates the internal and external elements that could affect the programme’s success.
Immunisation	A process by which a person becomes protected against a disease through vaccination or recovery from infection [11].
Monitoring	The systematic process of collecting, analysing, and using information to track progress toward objectives and guide management decisions [10].
M&E framework	A matrix compiling goal/purpose, outcomes, and outputs, along with the defined and measurable indicators with specified targets/thresholds necessary to achieve success.
Vaccination	The management and administration of vaccines pre-/peri-/post-vaccination to provide people with the most effective immunisation [11].
Vaccine	A product, usually administered through needle injection, by mouth, or sprayed into the nose, that stimulates a person’s immune system to produce immunity to a specific disease, protecting the person from that disease [11].

**Table 2 vaccines-10-00567-t002:** Eligibility criteria.

Criteria	Included	Excluded
1. Context	National settings	Other (e.g., subnational, international)
2. Topic	Vaccine implementation	Vaccine production Vaccine R&D
3. Outcomes	Framework Indicators Lessons Impact	Other
4. Source type	Primary research articles Review articles that include studies not included individually Commentaries/editorials if they include primary research Conference abstracts that include primary research Book chapters that include primary research	Audio/video reports Conference abstracts covering the same material as an available publication Social media, blogs, media articles Guidance/legal documents
5. Time-period	Any	NA
6. Language	All languages	NA
7. Study design	Any	NA
8. Participants	Any	NA

**Table 3 vaccines-10-00567-t003:** Synthesised findings by source.

Lead Author, Year	Type	Country/ies	Approach	M&E Framework	Coverage Indicators			Operational Indicators				Clinical Indicators		Lessons Learnt
					Targeting/Estimation	Equity	Uptake	Service Capacity	Vaccine Supply	Human Resources	M&E Costs	Vaccine Safety	Vaccine Demand	
Aceituno, 2017	Article	Bolivia	Quantitative	X			X					X		X
Al Awaidy, 2020	Article	multi (Bahrain, Kuwait, Oman, Qatar, Saudi Arabia, United Arab Emirates)	Quantitative	X			X							X
Alam, 2018	Article	Bangladesh	Quantitative				X							X
Ashish, 2017	Article	Nepal	Quantitative				X							X
Bawa, 2019	Article	Nigeria	Quantitative		X		X							X
Beard, 2015	Article	Australia	Quantitative		X		X					X		X
Bednarczyk, 2019	Article	US	Quantitative											X
Bernal, 2021	Article	UK	Quantitative											X
Bhatnagar, 2016	Article	India	Quantitative											X
Bianco, 2012	Abstract	Italy	Quantitative		X							X		
Carrico, 2014	Article	US	Mixed							X				X
Checchi, 2019	Article	UK	Quantitative		X		X							X
Cherif, 2018	Article	Ivory Coast	Quantitative							X				X
Cutts, 1988	Article	Mozambique	Quantitative											X
D’Ancona, 2018	Article	Italy	Quantitative		X									X
Dang, 2020	Article	Viet Nam	Quantitative	X										X
Edelstein, 2019	Article	UK	Quantitative				X							X
Geoghegan, 2021	Abstract	Ireland	Qualitative			X								
Hall, 2021	Article	UK	Quantitative		X					X				
Hipgrave, 2006	Article	multi (China, Indonesia, Viet Nam)	Quantitative									X		
Hutubessy, 2012	Article	Tanzania	Quantitative								X			X
Ijsselmuiden, 1987	Article	South Africa	Mixed				X							X
Imoukhuede, 2007	Article	Gambia	Quantitative											X
Lacapere, 2011	Article	Haiti	Quantitative		X		X							X
Lanata, 1990	Article	Peru	Quantitative				X							X
Loughlin, 2012	Article	US	Quantitative									X		X
Maina, 2017	Article	Kenya	Mixed				X							X
Manyazewal, 2018	Article	Ethiopia	Mixed	X	X		X	X	X	X				X
McCarthy, 2013	Article	US	Quantitative		X									X
Muhamad, 2018	Article	Malaysia	Quantitative				X							X
Ozdemir, 2010	Article	Turkey	Quantitative		X				X					X
Raji, 2019	Abstract	Nigeria	Qualitative											X
Richard, 2008	Article	Switzerland	Quantitative				X							X
Sarker, 2019	Article	Bangladesh	Quantitative			X								X
Soi, 2020	Article	multi (Bangladesh, Mozambique, Uganda, Zambia)	Quantitative	X										X
Tanton, 2017	Article	UK	Mixed		X									X
Tuells, 2010	Article	Spain	Qualitative											X
van Wijhe, 2018	Article	Netherlands	Quantitative		X		X							X
Vivekanandan, 2012	Article	India	Quantitative											X
Walker, 2014	Article	Kenya	Quantitative					X	X	X	X			X
Ward, 2017	Article	Uganda	Quantitative											X
Watson, 2010	Article	US	Quantitative											X
Wattiaux, 2016	Article	Australia	Mixed		X	X								X
			Totals	5	13	3	16	2	3	5	2	5	0	39

## Data Availability

Please contact the corresponding author for data requests.

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
