# Peer review of "Monitoring and Evaluation of National Vaccination Implementation: A Scoping Review of How Frameworks and Indicators Are Used in the Public Health Literature"

_vaccines, 2022, doi:10.3390/vaccines10040567_

Round 1

Reviewer 1 Report

In this review, the authors summarized the literature describing monitoring and evaluation (M&E) frame works or indicators for use in vaccination program implementation. The authors conducted a scoping review using Arksey and O’Malley’s six-stage framework and finally selected 43 eligible sources. This review identified literature on assessment frameworks and indicators used in national vaccination and integrated methods and lessons to inform the development of future frameworks. However, the introduction and figures of this review were insufficient and incomplete. The significance of the research was not explicitly stated in the manuscript. Judging from the above, I do not recommend this manuscript to be published at the present version.

Major points:

1. The introduction of the manuscript was not well-organized. It introduced the M&E frameworks, and the M&E for national COVID-19 vaccination program. However, there was no description of the necessity and importance of this review, resulting in an unclear objective of this review.

2. The abstract of the review was not completed. The analytical results of the literature included were illustrated, but the conclusion of the results was absent.

3. The results part “3.2. Synthesised findings” seemed to be the most important part of the review. The manuscript would be better to be understood if the authors use a graphical abstract to summarize the four major points.

4. The picture of figure 1 was not clear. Please replace it with a high-resolution version.

Minor points:

1. There are some abbreviations without references to explain the full names, such as OECD, SMS, MAPI.

Reviewer 2 Report

In this study, Marzouk and colleagues have developed a survey on monitoring and evaluation of national vaccination implementation. The authors used Arksey and O’Malley’s six-stage framework for the current scoping review, and then the findings were summarised thematically. The article looks fine and has the proper structure. Some recommendations for acceptance:

The figure's quality is not high, and the text is unclear within the figures. It is recommended that authors provide high-quality images for the article.

It is recommended that authors describe more on the discussion and adequately discuss the finding they found based on the articles reviewed. The discussion part of the article requires enrichment.

Providing a figure on previous studies and the field direction for future studies is recommended and increases the impact of the article.

Do the authors investigate the articles in the bioRxiv or similar websites? What is the status of those articles in the authors’ opinion?

Reviewer 3 Report

The paper aims at ‘evaluating the evaluations’ of vaccination from published literature. It reviewed 43 selected articles on monitoring and evaluation of national vaccination campaigns in countries around the world.

Needless to say that that paper is very partial, as it takes into account only the published literature, and ignores the large majority of evaluations made by government bodies, expert groups, pharmaceuticals, independent surveys, etc.

            The paper could still be useful for people involved in evaluation, as it presents serious conceptual frameworks, and it is well presented, well written and well organized.

Comments

1) Frameworks; what is often missing in these evaluations is the most important: the impact of the vaccination in the general population (demographic impact). This is most important for controversial vaccines such as vaccines against covid-19, malaria, cholera, influenza, etc., which have little demographic impact despite high vaccine coverage. This is a serious ‘gap’ for most evaluations, and indeed hard to do. A word on this point in the discussion would be most helpful.

2) Many evaluations are conducted by a variety of organizations outside the academic realm: government bodies (ministry of health), expert groups (working for the government or private groups), pharmaceuticals, independent surveys. A word on this point would be useful.

3) Some of the papers quoted do not have the aim to evaluate the full vaccination program, but only one aspect of it (cost, equity, personnel, vaccine technology, etc.). So, they cannot be compared.

4) The paper could better distinguish between routine data collection (on infectious diseases for instance), dedicated registers (focused on a disease, as rabies, tetanus, etc.), health statistics (morbidity, mortality), general surveys (as the demographic and health surveys in developing countries), and specific surveys for evaluation. All have strengths and weaknesses for evaluating vaccination.

5) The paper could better distinguish between quantitative and qualitative data. For instance, refusal of vaccination (such as for covid-19) needs to be investigated in a sociological / psychological framework. This type of analysis will appear in a totally different literature.

6) Authors should distinguish between the framework and the method for evaluation (‘register’, ‘survey’ versus ‘Lot Quality Assurance Sampling’ etc.).

7) Authors could have an annex table counting the number of occurrence of the main points addressed in the 43 articles [optional].

Details

Abstract; also line 56 etc. “Therefore, documentation of more experiences and lessons is needed to better inform M&E of vaccination beyond the second year of life threshold.”

            Why “second year of life”? Evaluation is needed for any vaccine, from birth (polio), early infancy to death (covid-19)

Line 142: Figure 3 shows 27 countries (not 29)

Line 176 and elsewhere: ‘Aceituno et al’s use’. The expression ‘et al.’ is abbreviation for a latin word: ‘et alii’ ; it cannot be used as ‘et al’s’.

Some typos remain in the reference list (ref. 1; font; etc.)

Round 2

Reviewer 1 Report

In this review, the authors summarized the literature describing monitoring and evaluation (M&E) frame works or indicators for use in vaccination program implementation. The authors conducted a scoping review using Arksey and O’Malley’s six-stage framework and finally selected 43 eligible sources. This review identified literature on assessment frameworks and indicators used in national vaccination and integrated methods and lessons to inform the development of future frameworks. Although the manuscript has some practical significance and the elements of a review article were included, the figures of this review were not qualified. Judging from the above, I do not recommend this manuscript to be published at the present version.

Major points:

  1. The picture of figure 1 is still not clear. Please replace it with a high-resolution version.
  2. The histograms of figure 2 and figure 3 are not well-designed. Please draw them again to highlight the key points of the results.

Reviewer 2 Report

Authors address my concerns properly 

Author Response

Dear Editors,

Thank you, we really appreciate your comment. We have uploaded new figures in better quality to address the reviewer comment. 

Sincerely,

Study authors

Round 3

Reviewer 1 Report

In this review, the authors summarized the literature describing M&E frame works or indicators for use in vaccination program implementation. The authors conducted a scoping review using Arksey and O’Malley’s six-stage framework and finally selected 43 eligible sources. This review identified literature on assessment frameworks and indicators used in national vaccination and integrated methods and lessons to inform the development of future frameworks. The authors had revised the manuscript according to previous comments. However, the manuscript was recommended to get specific editing by professionals before publication.

Major points:

  1. The English language should get editing by professionals. Some parts of the manuscript were encouraged to be elucidated in an objective way.
